# Sequence Model Imitation Learning with Unobserved Contexts

**Gokul Swamy**
Carnegie Mellon University
gswamy@cmu.edu

**Sanjiban Choudhury**
Cornell University
sanjibanc@cornell.edu

**J. Andrew Bagnell**
Aurora Innovation and Carnegie Mellon University
dbagnell@ri.cmu.edu

**Zhiwei Steven Wu**
Carnegie Mellon University
zstevenwu@cmu.edu

## Abstract

We consider imitation learning problems where the learner's ability to mimic the expert increases throughout the course of an episode as more information is revealed. One example of this is when the expert has access to privileged information: while the learner might not be able to accurately reproduce expert behavior early on in an episode, by considering the entire history of states and actions, they might be able to eventually identify the hidden context and act as the expert would. We prove that on-policy imitation learning algorithms (with or without access to a queryable expert) are better equipped to handle these sorts of *asymptotically realizable* problems than off-policy methods. This is because on-policy algorithms provably learn to recover from their initially suboptimal actions, while off-policy methods treat their suboptimal past actions as though they came from the expert. This often manifests as a *latching* behavior: a naive repetition of past actions. We conduct experiments in a toy bandit domain that show that there exist sharp *phase transitions* of whether off-policy approaches are able to match expert performance asymptotically, in contrast to the uniformly good performance of on-policy approaches. We demonstrate that on several continuous control tasks, on-policy approaches are able to use history to identify the context while off-policy approaches actually perform *worse* when given access to history.

## 1 Introduction

An unstated assumption in much of the work in imitation learning (IL) is that the learner and the expert have access to the same state information. With powerful enough statistical models, this assumption places us in the *realizable* setting – i.e. the imitator can actually behave like the expert. In practice however, an expert might have more information than the learner does. For example, an expert policy might be trained in simulation with privileged access to state before being used to supervise a learner policy that only has access to a subset of features. This recipe has enjoyed success in domains from motion planning with obstacles [Choudhury et al., 2018], to autonomous driving [Chen et al., 2019], to legged locomotion [Lee et al., 2020, Kumar et al., 2021].

Recent theoretical work has established "no-go" results for successfully imitating an expert that has access to more information [Zhang et al., 2020, Kumor et al., 2021]. The core of their arguments is that without seeing some feature that influences expert behavior but is not echoed elsewhere in the state, the learner might not be able to properly ground the expert actions in the observed state. In causal inference terms, this hidden information acts as an *unobserved confounder* which prevents identification of the desired causal estimand (the expert action). Despite these results,

36th Conference on Neural Information Processing Systems (NeurIPS 2022).

impressive empirical successes have been achieved even when the learner has a more impoverished state representation than the expert.

In this work, we reconcile theory and practice by considering a broad class of problems where the learner's ability to mimic expert actions increases as more observations are revealed. We study the large-horizon limit to tease out what is key to good performance. We find that off-policy approaches (e.g. behavioral cloning) that ignore the resulting covariate shift from initially sub-optimal decisions can lead to poor results, even when there exists a policy that in the large-horizon limit is optimal. We show that for some problem families, there exists a sharp *phase transition* in problem parameters where off-policy IL shifts between being consistent to having arbitrarily poor performance.

In contrast, we show that on-policy approaches that leverage interaction with the demonstrator [Ross et al., 2010] or take advantage of interaction with the environment (in the style of Inverse Optimal Control [Bagnell, 2015]) [Ziebart et al., 2008, Ho and Ermon, 2016] are always (i.e. independent of parameters) asymptotically consistent on these problems. We believe this strong separation between on- and off- policy approaches helps explain both the poor performance of behavioral cloning even in regimes with large data and powerful model classes [Spencer et al., 2021, Muller et al., 2006, Codevilla et al., 2019, de Haan et al., 2019, Bansal et al., 2018, Kuefler et al., 2017] and the success of on-policy methods mentioned above.

We note that, in contrast to the hidden state that is common in real-world problems [Boots et al., 2011, Kumar et al., 2021, Lee et al., 2020], standard benchmarks like the PyBullet suite [Coumans and Bai, 2016] are fully observed, enabling off-policy algorithms like behavioral cloning to match expert performance [Swamy et al., 2021]. Thus, for our experiments, we introduce partial observability to ensure that we are focused on part of what makes imitation learning hard in practice.

We study in detail Contextual Markov Decision Process (CMDPs) [Hallak et al., 2015] that satisfy an *asymptotic realizability* condition. Intuitively, this means we can expect that proper utilization of history to eventually enable accurate prediction of the context. A key result we show is that ***for identifiable CMDPs where the learner can recover from mistakes early on in an episode, on-policy imitation learning algorithms that operate in the space of histories are able to asymptotically match time-averaged expert value, while off-policy approaches struggle to do so.***

More concretely, our work makes three contributions:

1. We show that under appropriate identifiability and recoverability conditions, the context-dependent expert policy becomes *asymptotically realizable*, enabling on-policy imitation learning algorithms to match (or nearly match) time-averaged expert performance.

2. More generally, we show that when longer history allows the learner to get closer to realizing the expert policy, on-policy methods are able to take advantage of this property while off-policy methods are stuck with the consequences of their mistakes early on in an episode. This manifests as off-policy methods producing policies that merely repeat previous actions.

3. We conduct experiments in a simplified bandit domain which show that there exist sharp *phase transitions* in terms of when off-policy imitation learning algorithms match expert performance in contrast to the uniform value-equivalence of the policies produced by on-policy approaches. We also conduct experiments on continuous control tasks that show that on-policy algorithms are able to take advantage of history to correctly identify the context in a way that off-policy methods are not.

We begin with a discussion of related work.

## 2 Related Work

One of the fundamental challenges of imitation learning (or any sequential prediction task where the learner consumes some function of its own prior predictions) is the likelihood of significant covariate shift between training-time data and test-time observations [Ross and Bagnell, 2010]. In short, this happens because the learner might end up in states not seen in the demonstrations and is thus unsure how to act. Early work in this area includes that of Daumé et al. [2009] in the natural language processing and that of Ross et al. [2010] in imitation learning and robotics, both of which come to the preceding conclusion. Recent work by Spencer et al. [2021] shows that there are actually more than one regime of covariate shift. In the "easy" regime where the expert is realizable, off-policy

methods like behavior cloning match expert performance when data and model capacity are large enough. However, in harder regimes where the expert is non-realizable due to model misspecification, off-policy methods compound in error and one must rely on on-policy methods, either those that require an interactive expert [Ross et al., 2010] or an interactive simulator [Ziebart et al., 2008, Swamy et al., 2021].

One instance of model misspecification of practical interest is when the learner is denied state information that the expert uses. For instance, in self-driving, the human expert might have richer context about the scene than the limited perception system of the car. While the standard solution is to add a history of past states and actions to the model, practitioners have often noted that this leads to a "latching effect" where the learner simply repeats the past action. For example, Muller et al. [2006] note such latching with steering actions, Kuefler et al. [2017], Bansal et al. [2018] note this with braking actions, and Codevilla et al. [2019] with accelerations. Recent work by Ortega et al. [2021] also points out this latching behavior which they term as "self-delusion."

Once one identifies the downstream effect of missing context as covariate shift, a natural question is whether an extension of prior covariate-shift-robust imitation learning methods to the space of histories would be able to learn effectively in partial information settings. Prior work has answered parts of this question. For example, Choudhury et al. [2018] proves that interactive imitation learning over the space of histories converges to the QMDP approximation of the expert's policy [Littman et al., 1995]. Recent work Ortega et al. [2021] views these kind of partial information setting through a causal lens [Pearl et al., 2016], and provides an algorithm (counterfactual teaching) equivalent to the interactive-expert FORWARD algorithm of Ross and Bagnell [2010] under log-loss. We build upon these analyses by providing conditions under which such approaches will converge to a policy that is equivalent in value to that of the expert in the presence of unobserved contexts.

Our focus on the contextual MDP [Hallak et al., 2015] is because, as argued by [Zhang et al., 2020, Kumor et al., 2021], context that is updated throughout the episode and is only reflected for a single step prevents the learner from refining its estimates via considering history. Our results apply to settings beyond the CMDP where a similar asymptotic realizability property holds. Tennenholtz et al. [2021] consider IL in contextual MDPs but give the learner access to the confounder at test time and do not consider policies that operate over the space of histories. de Haan et al. [2019] also consider imitation learning from a causal inference perspective but focus on issues of covariate shift [Spencer et al., 2021], rather than those of missing information. [Swamy et al., 2022] also consider unobserved confounders in IL but focus on correlated action perturbations rather than partial observability.

Wen et al. [2020, 2021, 2022], Chuang et al. [2022] propose various *offline* approaches to mitigating the latching effect (which they term the copycat problem). However, as we prove below, *online* interaction with the environment is both necessary and sufficient to prevent compounding errors and the associated latching effect. It is therefore difficult to provide strong theoretical guarantees for their work or argue that their results in simulation would necessarily transfer to real-world problems.

## 3   The Latching Effect in Off-Policy Imitation Learning

Consider a finite-horizon Contextual Markov Decision Process (CMDP) parameterized by $\langle \mathcal{S}, \mathcal{A}, \mathcal{C}, \mathcal{T}, r, T \rangle$ where $\mathcal{S}, \mathcal{A}, \mathcal{C}$ are the state, action, and context spaces, $\mathcal{T} : \mathcal{S} \times \mathcal{A} \times \mathcal{C} \to \Delta(\mathcal{S})$ is the transition operator, $r : \mathcal{S} \times \mathcal{A} \times \mathcal{C} \to [-1, 1]$ is the reward function, and $T$ is the horizon. At the beginning of each episode, a context is sampled from $p(c)$ and is held fixed until the next reset. Intuitively, there exists a family of reward and transition functions that are indexed by the context for each episode. We use $h_t \in \mathcal{H}$ to denote a $t$-step history: $(s_1, a_1, \ldots, s_t)$. We see trajectories generated by an expert policy $\pi^E : \mathcal{S} \times \mathcal{C} \to \Delta(\mathcal{A})$ and search over time-varying policies $\pi : \mathcal{H} \to \Delta(\mathcal{A})$. We use $\pi_{1\ldots t}$ to refer to the sequence of policies that comprise this time-varying policy. We assume that our policy class $\Pi$ is convex and compact.

We begin with a toy example of the *latching effect* and how it leads to poor policy performance.

**Problem 3.1** (Causal Bandit Problem, [Ortega et al., 2021])**.** Consider an episodic MDP with $K$ actions (arms) and a single state. At the beginning of each episode, a context $c \in [K]$ is chosen uniformly at random and represents the correct arm for that episode. Pulling an arm leads to binary feedback: $+$ if the arm was correct and $-$ otherwise. This feedback is flipped with probability $\epsilon_{obs} \in [0, 1]$. The learner observes expert demonstrations where at each timestep, the expert plays the correct arm with probability $1 - \epsilon_{exp} \in [0, 1]$ and another arm uniformly at random otherwise. The

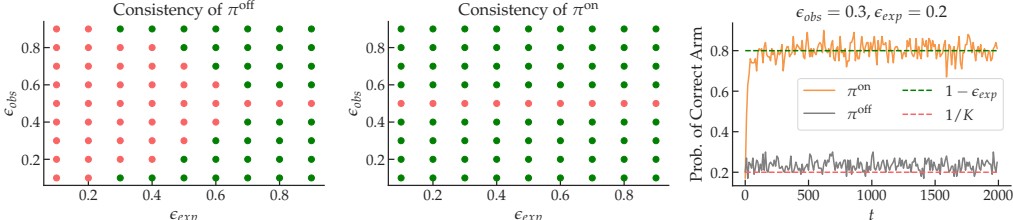

Figure 1: We plot whether the learner's policy has a higher probability of making mistakes than the expert ($\epsilon_{eps}$) after 2000 steps, averaged across 100 trials on instances with $K = 5$. We use red dots to indicate when this is true and green dots otherwise. We see the on-policy method match expert performance everywhere the problem is identifiable in contrast to the off-policy method, for which a slight perturbation of a parameter can lead to a drastically different result in terms of long-term performance. On a particular problem setting, we see the off-policy method pick a random arm and repeat it ad infinitum and therefore perform at the level of random chance while the on-policy method is able to match expert performance.

reward function is 1 for pulling the correct arm and 0 otherwise. We emphasize that the learner does not observe the rewards, just noisy binary feedback as an observation after each pull.

As the learner does not observe the correct arm, we are in the partial information setting. However, as one might expect, by pulling all arms enough times and observing the noisy feedback, the learner can narrow down which arm they should pull for the rest of the episode. Note that the learner stays in the same state after each action so they are free to perform this exploration without long-term consequences.

**Observation 1: Off-Policy Methods Have Consistency Phase Transitions.** In Fig. 1, we plot, for a variety of settings of the problem parameters ($\epsilon_{exp}, \epsilon_{obs}$) whether the learner makes mistakes with a significantly higher probability than the expert does for an off-policy IL algorithm and an on-policy IL algorithm. We give all learners access to the full history of interactions (i.e. arms pulled and noisy feedback observed). With exactly $\epsilon_{obs} = 0.5$, the learner gets no information from the observations, preventing any algorithm from learning properly. This means that under this particular setting, the problem instance is not *identifiable*, a concept we develop further below. In contrast to the uniform consistency of the on-policy approach whenever the problem is identifiable, we see sharp *phase transitions* in terms of where the off-policy approach is consistent – a small change to either problem parameter can lead to a drastically different result.

**Observation 2: Off-Policy Methods Produce Latching Policies.** At the first timestep, all learners pick an arm uniformly at random as they have no information about the context. In the cases where the off-policy approach is inconsistent, it continues to pick this arm *ad infinitum* as it treats its own past actions as though they were the expert's. This is what leads to the $\frac{1}{K} = \frac{1}{5}$ success rate seen in the rightmost part of Fig. 1, even after 2000 timesteps of experience telling the learner that another arm should be pulled. Put differently, the off-policy learner collapses its uncertainty over the correct arm too quickly for the negative feedback it receives to push it to a different arm. Concerningly, this effect appears to be extremely sensitive to the parameters of the problem, rendering it difficult to predict and hedge against.

**Observation 3: Low Density Ratios Help Off-Policy Methods.** Looking at the left side of Fig. 1, a natural question might be why, as we increase $\epsilon_{exp}$, the off-policy approach is consistent for a wider spectrum of $\epsilon_{obs}$ values. Observe that as we increase $\epsilon_{exp}$, the expert has a higher chance of making a mistake, bringing its trajectory distribution closer to that of the uniform policy and lowering the maximum density ratio between learner and expert trajectory distributions. As was established by Spencer et al. [2021], a low maximum density ratio is a sufficient condition for behavioral cloning to be consistent. This experiment appears to echo their theoretical results.

Putting together Fig. 1, on-policy approaches appear to be able to handle hidden context given access to history while off-policy can do so only in a way that depends heavily on problem parameters.

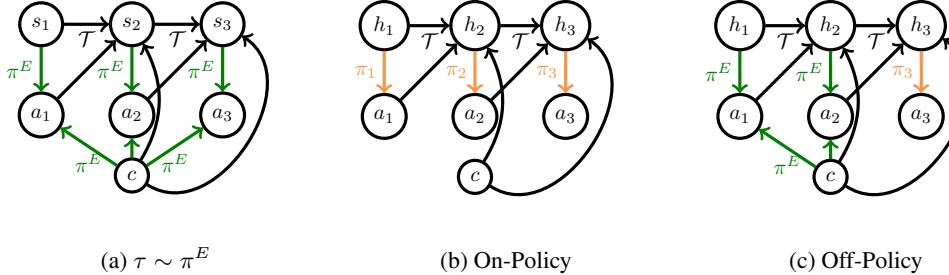

(a) $\tau \sim \pi^E$  (b) On-Policy  (c) Off-Policy

Figure 2: **(a)**: The SCM that corresponds to the generative process for expert trajectories. **(b)**: The SCM corresponds to the generative process for learner trajectories in reality. **(c)**: The SCM that corresponds to the generative process that off-policy algorithms assume – intuitively, it corresponds to the expert taking all actions up till the current timestep and then handing off control.

# 4 A Bayesian Perspective on the Latching Effect

To begin to explain this difference in behavior of on-policy and off-policy algorithms, we consider the structural causal model (SCM) each of these classes of algorithms is implicitly assuming.

## 4.1 SCMs for Imitation Learning

We observe that expert trajectories of length $t$ are generated according to

$$p(\tau; \pi^E) \triangleq p(c)p(s_1) \prod_{i=1}^{t-1} \pi^E(a_i|c, s_i)\mathcal{T}(s_{i+1}|s_i, a_i, c), \tag{1}$$

while learner trajectories are generated according to

$$p(\tau; \pi) \triangleq p(c)p(s_1) \prod_{i=1}^{t-1} \pi_t(a_i|h_i)\mathcal{T}(s_{i+1}|s_i, a_i, c). \tag{2}$$

We use $\tau \sim \pi^E$ and $\tau \sim \pi$ to denote $T$-step trajectories sampled according to the above distributions. We define our value and Q functions as usual: $V^\pi(s) = \mathbb{E}_{\tau \sim \pi|s_t=s}[\sum_{t'=1}^{T} r(s_{t'}, a_{t'})]$, $Q^\pi(s, a) = \mathbb{E}_{\tau \sim \pi|s_t=s,a_t=a}[\sum_{t'=1}^{T} r(s_{t'}, a_{t'})]$. Lastly, let performance be $J(\pi) = \mathbb{E}_{\tau \sim \pi}[\sum_{t=1}^{T} r(s_t, a_t)]$.

The key difference between on-policy and off-policy imitation learning algorithms is the difference between the center and right SCMs of Fig. 2. On-policy algorithms assume the data they have seen thus far is generated by executing learner $\pi$ (b) while off-policy algorithms assume it is generated by executing the expert policy $\pi^E$ (c). The off-policy approximation is what leads to the latching behavior observed empirically: even if the first action was chosen at random, *by treating it as though it was produced by the expert $\pi^E$ (who sees the context c and therefore picks actions that are correlated across time), the learner is likely to continue to repeat the past action.* This is also what we observe empirically in the causal bandit problem in the cases where off-policy IL does not work: the learner continues to play the first arm it chose, ignoring the feedback it gets that it is repeatedly making a mistake, and only matching expert performance on a $\frac{1}{K}$ of episodes. We note this gives the off-policy learner a choice between fire and pyre (in the words of Eliot [1942]): they need to use history to narrow down the context but if they do, they can learn a latching policy that performs poorly at test-time.

## 4.2 Off Policy Methods Have an Incorrect Context Posterior

Another way of seeing this point is by considering what the posterior over the confounder would be under samples from each of these SCMs. If the learner was able to accurately pin down the correct arm, they would be able to easily reproduce the expert policy. Thus, we can focus on correct-arm identification via Bayes Rule. Assuming a uniform prior over contexts,

$$p(c|h_t) \propto p(c, h_t). \tag{3}$$

Under the off-policy graphical model, $p(c, h_t) \propto p(\tau; \pi^E)$: the probability of a history $h_t$ under the expert's distribution (Fig. 2, (a)). Expanding terms, we arrive at the following expression

**Proposition 4.1.** *The off-policy posterior over contexts is*

$$p_{\text{off}}(c, h_t) \propto p(h_t; \pi^E) \propto p(c)p(s_1) \prod_{i=1}^{t-1} \pi^E(a_i|c, s_i)\mathcal{T}(s_{i+1}|s_i, a_i, c). \qquad (4)$$

Notice that for the expert policy, the context $c$ directly influences actions so conditioning on actions when attempting to predict $c$ is correct. We highlight in green the term that encodes this dependence. In contrast, if we assume the learner has no access to the context except via its influence on the history of states, we should instead treat actions as *interventions* [Pearl et al., 2016] that provide no further information about the context. The on-policy graphical model (Fig. 2, (b)) does just this, allowing us to compute the correct posterior over contexts. Formally,

**Proposition 4.2.** *The on-policy posterior over contexts is*

$$p_{\text{on}}(c, h_t) \propto p(h_t; \pi) = p(c|do(a_1)\dots do(a_{t-1}), s_1\dots s_t) \propto p(c)p(s_1) \prod_{i=1}^{t-1} \mathcal{T}(s_{i+1}|s_i, a_i, c), \quad (5)$$

where the equality follows from the fact that

$$(c \perp a_{1\dots t}|s_{1\dots t})_{\mathcal{G}_{\underline{a_{1:t}}}} \qquad (6)$$

and the standard $do()$-calculus rules [Pearl et al., 2016]. Intuitively, we are leveraging the fact that the learner's actions have all their dependence on the context mediated through the history of states to ignore them in our posterior calculation. Notice that this expression matches the off-policy posterior except for the term in green: *because the on-policy learner knows that the actions in the history were not produced by the expert, it does not weight them by the expert's probability of playing them in its posterior update.* Graphically, as far as the posterior over the context is concerned, on-policy approaches correspond to the SCM in Fig. 3, left.

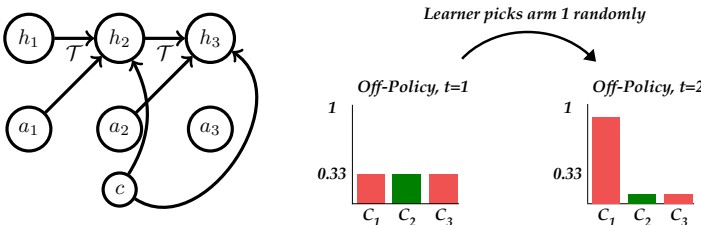

Figure 3: **Left:** The SCM for a CMDP that treats actions as interventions that provide no useful information about the context. This SCM is what the on-policy learner uses when performing posterior updates. **Right:** By treating its own actions as though they came from the expert, an off-policy learner's posterior collapses too quickly on an incorrect context, causing latching.

By ignoring its own incorrect actions early on, the on-policy learner doesn't over-index on them and come to a false conclusion as to what the value of the context is, as illustrated in Fig. 3, right. We've already seen evidence of how the addition or removal of the green term can lead to markedly different results: the learners we used to generate Fig. 1 had policies given by

$$\pi^{\text{off}}(a_t|h_t) = \sum_{c \in \mathcal{C}} p_{\text{off}}(c|h_t)\pi^E(a_t|c, s_t), \qquad (7)$$

and

$$\pi^{\text{on}}(a_t|h_t) = \sum_{c \in \mathcal{C}} p_{\text{on}}(c|h_t)\pi^E(a_t|c, s_t). \qquad (8)$$

Note that these policies are exactly equivalent except for the green term in the posterior of the off-policy $\pi^{\text{off}}$, yet respond quite differently as the parameters of the problem are changed. Also note that $p_{\text{on}}(c|h_t)$ requires interaction with the environment to evaluate, while $p_{\text{off}}(c|h_t)$ does not.

# 5    A Moment-Matching Solution to the Latching Effect

We now turn our attention to generalizing this argument beyond Bayesian learners, including to those that match sufficient statistics of expert behavior – *moments* – rather than maintaining explicit posteriors, and providing value equivalence guarantees for history-equipped on-policy learners.

## 5.1    A Quick Review of Moment-Matching in Imitation Learning

First, as in Swamy et al. [2021], we define $\mathcal{F}_{Q_E}$ as the set of *on-Q moments*, with $f \in \mathcal{F}_{Q_E}$ satisfying type signature $\mathcal{S} \times \mathcal{A} \times \mathcal{C} \to [-T, T]$. Intuitively, $\mathcal{F}_{Q_E}$ spans the set of possible expert $Q$-functions. We require the actual expert $Q$-function to be contained: $Q^{\pi_E} \in \mathcal{F}_{Q_E}$. We assume that $\mathcal{F}_{Q_E}$ is convex, compact, closed under negation, and finite dimensional. Second, define $\mathcal{F}_Q$ be the class of *off-Q moments* (i.e. $\forall \pi \in \Pi$, $Q^\pi \in \mathcal{F}_Q$) and satisfy the same function-class assumptions as $\mathcal{F}_{Q_E}$. Third, let $\mathcal{F}_r$ denote the class of reward moments and also satisfy the same function-class assumptions. All $f \in \mathcal{F}_r$ satisfy type signature $\mathcal{S} \times \mathcal{A} \times \mathcal{C} \to [-1, 1]$ and we assume that $r \in \mathcal{F}_r$.

Swamy et al. [2021] prove that if one is able to approximately solve a two-player zero-sum *moment-matching game* between a policy player (that picks from $\Pi$) and a discriminator (that picks from $\mathcal{F}_{Q_E}$, $\mathcal{F}_Q$, or $\mathcal{F}_r$), one has a bound on the performance difference between the learner and the expert. Depending on which class the discriminator selects from, one ends up with a different bound. For example, if one solves the game with the following payoff and $f \in \mathcal{F}_r$ to an $\epsilon$-approximate Nash equilibrium,

$$U(\pi, f) = \frac{1}{T}\left(\mathop{\mathbb{E}}_{\tau \sim \pi}\left[\sum_{t=1}^T f(s_t, a_t, c)\right] - \mathop{\mathbb{E}}_{\tau \sim \pi^E}\left[\sum_{t=1}^T f(s_t, a, c)\right]\right), \tag{9}$$

one has a guarantee that $J(\pi^E) - J(\pi) \le \epsilon T$. Unfortunately, directly solving such a game is not possible in the hidden context setting as we do not see the contexts and therefore cannot evaluate the moment functions. [1] We turn our attention to adapting moment-matching to the contextual setting.

## 5.2    Moment-Matching with Unobserved Contexts

For each moment class, we assume the existence of an *observable moment class* that operates over the space of histories (i.e. $\mathcal{H} \times \mathcal{A} \to \mathbb{R}$) and has members that eventually produce outputs close to that of their context-dependent counterparts. In math,

**Assumption 5.1** (Asymptotic On-$Q$ Moment Identifiability)**.**

$$\forall f \in \mathcal{F}_{Q_E}, \forall c \in \mathcal{C}, \exists \tilde{f} \in \tilde{\mathcal{F}}_{Q_E} \text{ s.t. } \lim_{T \to \infty} \sup_{\substack{\pi \in \Pi \cup \{\pi^E\} \\ a \in \mathcal{A}}} \mathbb{E}_{\tau \sim \pi, c}[f(s_T, a, c) - \tilde{f}(h_T, a)] = 0, \tag{10}$$

Analogous conditions can be defined for $\mathcal{F}_r$ and $\mathcal{F}_Q$, giving us $\tilde{\mathcal{F}}_r$ and $\tilde{\mathcal{F}}_Q$. We note that these conditions are asymptotic in nature – we use $\delta(t)$ to refer to the finite-horizon expected difference between the observable and context-dependent moments. We use $H$ to denote the *moment recoverability constant*, which bounds how much total cost is incurred for the expert to recover from an arbitrary mistake [Swamy et al., 2021]:

$$H = \left|\sup_{\substack{a \in \mathcal{A}, s \in \mathcal{S} \\ c \in \mathcal{C}, f \in \mathcal{F}_{Q_E}}} f(s, a, c) - \mathbb{E}_{a' \sim \pi^E(s, c)}[f(s, a', c)]\right| \tag{11}$$

For problems where the expert is able to effectively correct learner mistakes, this quantity can be significantly smaller than the horizon. Define $\tilde{\mathcal{F}}_{\text{on}} = \{\tilde{f}/2H : \tilde{f} \in \tilde{\mathcal{F}}_{Q_E}\}$ and $\tilde{\mathcal{F}}_{\text{off}} = \{\tilde{f}/2T : \tilde{f} \in \tilde{\mathcal{F}}_Q\}$ to be scaled-down versions of the observable moments such that their range is $[-1, 1]$. We can now define our moment-matching errors:

$$\epsilon_{\text{on}}(t) = \sup_{\tilde{f} \in \tilde{\mathcal{F}}_{\text{on}}} \mathbb{E}_{\tau \sim \pi}[\tilde{f}(h_t, a_t) - \mathbb{E}_{a' \sim \pi^E(s_t, c)}[\tilde{f}(h_t, a')]], \tag{12}$$

---

[1]The astute reader might notice that we need access to the context to *simulate* learner rollouts as the context also affects the transition model. Thus, as long as one can simulate, one can do standard moment-matching in the space of context-dependent moments. We write things in history-space to also handle the real-world setting where contexts are always unavailable.

$$\epsilon_{\text{off}}(t) = \sup_{\tilde{f} \in \tilde{\mathcal{F}}_{\text{off}}} \mathbb{E}_{\tau \sim \pi^E}[\tilde{f}(h_t, a_t) - \mathbb{E}_{a' \sim \pi(h_t)}[\tilde{f}(h_t, a')]], \tag{13}$$

$$\epsilon_{\text{rew}}(t) = \sup_{\tilde{f} \in \tilde{\mathcal{F}}_{\text{r}}} \mathbb{E}_{\tau \sim \pi}[\tilde{f}(h_t, a_t)] - \mathbb{E}_{\tau \sim \pi_E}[\tilde{f}(h_t, a_t)]. \tag{14}$$

As argued by Swamy et al. [2021], $\epsilon_{\text{on}}(t)$ governs the performance of on-$Q$ algorithms like DAgger [Ross et al., 2010], $\epsilon_{\text{off}}(t)$ the performance of off-$Q$ algorithms like behavioral cloning [Pomerleau, 1989], and $\epsilon_{\text{rew}}(t)$ the performance of reward-matching algorithms like MaxEnt IRL [Ziebart et al., 2008] and GAIL [Ho and Ermon, 2016]. With these definitions laid out, we can now prove how well each class of algorithms handles unobserved contexts.

### 5.3 Asymptotic Realizability in Imitation Learning

For some partial information problems, the use of history coupled with on-policy feedback might allow the learner to eventually match expert performance. This is an example of a more general phenomenon we term *asymptotic realizability* (AR), in which the learner is able to perform as well as the expert does with high probability after observing an arbitrary history of some length. We begin by defining the *average imitation gap* or AIG for short.

**Definition 5.2** (AIG($\pi$, T))**.**

$$\mathbb{E}_{\tau \sim \pi^E}\Big[\frac{1}{T}\sum_{t=1}^{T} r(s_t, a_t, c)\Big] - \mathbb{E}_{\tau \sim \pi}\Big[\frac{1}{T}\sum_{t=1}^{T} r(s_t, a_t, c)\Big]. \tag{15}$$

We now define our performance target in AR problems.

**Definition 5.3** (Asymptotic Value Equivalence (AVE))**.** We say that policy $\pi$ is asymptotically value-equivalent (AVE) when the following condition holds true:

$$\lim_{T \to \infty} \text{AIG}(\pi, T) = 0. \tag{16}$$

Intuitively, this condition means that the learner performs as well as the expert does on average, given enough time. Put differently, we do not penalize the learner for initial mistakes as long as they are able to learn enough from them to match expert performance. We will proceed by studying the AIG properties of on-policy and off-policy imitation learning algorithms.

We are now ready to state our main result:

**Theorem 5.4** (AVE of IL Algorithms)**.** *Define* $\limsup_{t \to \infty} \epsilon_{\text{on}}(t) = \epsilon_{\text{on}}(\infty)$, $\lim_{T \to \infty} \sum_t^T \epsilon_{\text{off}}(t) + \delta_{\text{off}}(t) = \Sigma_{\text{off}}(\infty)$, *and* $\limsup_{t \to \infty} \epsilon_{\text{rew}}(t) = \epsilon_{\text{rew}}(\infty)$. *For all (C)MDPs and $\pi$,*

$$\lim_{T \to \infty} \text{AIG}(\pi, T) \leq \epsilon_{\text{on}}(\infty)H, \tag{17}$$

$$\lim_{T \to \infty} \text{AIG}(\pi, T) \leq \Sigma_{\text{off}}(\infty), \tag{18}$$

$$\lim_{T \to \infty} \text{AIG}(\pi, T) \leq \epsilon_{\text{rew}}(\infty). \tag{19}$$

In words, if either an on-$Q$ or reward-matching learner is able to achieve 0 moment matching error asymptotically, they will achieve the same average value as the expert. Even if they are unable to do so, the AIG will be bounded by a constant. In contrast, the result for off-policy algorithms is much weaker. Notice that instead of taking a $\limsup$ which, roughly speaking, captures the asymptotic error, the off-policy bound is in terms of the *sum* of errors, which factors in errors made early on. This result indicates that an off-policy learner that makes mistakes early on in the episode (as is likely for contextual problems) might be unable to eventually match expert performance, even when the problem is recoverable. We prove that there exist certain problems for which this bound is tight.

**Theorem 5.5** (Off-Policy AVE Lower Bound)**.** *There exist CMDPs and $\pi$ s.t.* $\limsup_{t \to \infty} \epsilon_{\text{off}}(t) = 0$ *for which*

$$\lim_{T \to \infty} \text{AIG}(\pi, T) \gtrsim \Sigma_{\text{off}}(\infty). \tag{20}$$

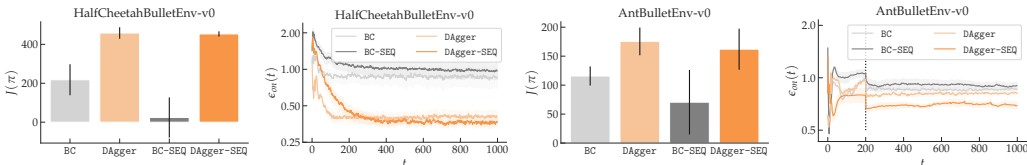

Figure 4: We use the suffix `-SEQ` to refer to models that have access to history (of length 5 for all experiments). Standard errors are computed across 4 runs. **Left:** We consider a modification of the HalfCheetah task where the goal is for the agent to run at a particular velocity. The expert sees this velocity while the learner observes an indicator of whether their current velocity is above or below the target, achieving $J(\pi_E) = 560$. We see that adding history to `BC` actually *reduces* the performance of the learned policies, in contrast, to `DAgger`. We also see that `DAgger-SEQ` eventually out-perform `DAgger` in terms of moment-matching error. **Right:** We consider a modification of the Ant task where the target velocity is only revealed to the learner at $t = 200$. The expert achieves $J(\pi_E) = 300$. We again see using sequence models harms `BC` performance while reducing `DAgger` moment-matching error. While all methods drop in error at $t = 200$, the drop is particularly large for the on-policy methods, indicating that they are better able to manage uncertainty over the context.

In words, this theorem says that for certain problems, *even if an off-policy learner can drive down error asymptotically, they might be doomed as far as AVE because of mistakes made early on.*

We prove these results in Appendix A. To come full circle, we now argue that CMDPs (including our causal bandit problem) that satisfy the asymptotic moment identifiability conditions along with an *asymptotic realizability* condition on the policy class enable the learner to match expert performance:

**Assumption 5.6** (Asymptotic Realizability). Asymptotic Moment Identifiability holds and $\exists \pi \in \Pi$ s.t. $\lim_{t \to \infty} \epsilon_{\text{on}}(t) = 0$, $\lim_{t \to \infty} \epsilon_{\text{off}}(t) = 0$, and $\lim_{t \to \infty} \epsilon_{\text{rew}}(t) = 0$.

Note this assumption is weaker than a standard realizability assumption – it is saying that along certain moments of interest, we can asymptotically match the expert. Plugging in this assumption into Theorem 5.4 tells us that on such problems, an on-policy learner will be able to achieve AVE.

We now turn our attention to efficiently computing such a policy. We prove that by solving an approximate equilibrium computation game over the space of history-based policies and the space of observable moments (which can be done efficiently with no-regret algorithms, [Freund and Schapire, 1997]), one can find a policy that achieves a low AIG.

**Theorem 5.7.** *For any contextual MDP and policy class that satisfies Asymptotic Realizability, let $\pi$ be an $\epsilon$-approximate Nash equilibrium strategy for the infinite horizon reward-matching or on-Q-matching game. Then, we know that $\lim_{T \to \infty} \text{AIG}(\pi, T) \leq \epsilon$ or $\lim_{T \to \infty} \text{AIG}(\pi, T) \leq H\epsilon$.*

In short, by solving an on-policy moment-matching problem over policies that have access to history, we have strong guarantees of matching expert performance on asymptotically realizable problems. We re-iterate that we have no such guarantees for off-policy algorithms. We also note that our causal bandit problem satisfies these assumptions, providing theoretical justification for our results.

**Corollary 5.8.** *There exists a singleton observable moment class such that the Causal Bandit Problem satisfies Asymptotic Realizability, implying that the iterates produced by an on-policy moment matching algorithm will achieve AVE.*

Putting it all together, for asymptotically realizable problems, on-policy imitation learning algorithms have stronger guarantees than their off-policy counterparts with respect to asymptotic value equivalence. If a contextual MDP satisfies these conditions (like with our Causal Bandit Problem), the preceding results apply. We see that our theory matches our experiments: on-policy algorithms appear to work on all identifiable instances of the causal bandit problems, while off-policy algorithms have much weaker performance. We now turn our attention to more complex CMDPs that satisfy our assumptions to further validate our theory empirically.

# 6 Experiments

We conduct experiments in a CMDP extension of the standard PyBullet tasks Coumans and Bai [2016] that is inspired by the multi-task reinforcement learning setups of Finn et al. [2017]. In these tasks, the agent is rewarded for running at a particular velocity that is randomly sampled at the beginning of each episode. We train expert policies that have access to this privileged information. We compare two algorithms: the off-policy behavioral cloning (`BC`) [Pomerleau, 1989] and the on-policy DAgger (`DAgger`) [Ross et al., 2010] that either have access to the immediate state or the last five timesteps of history (for which we use the suffix `-SEQ`).

In the bar plots of Fig. 4, we see that without access to history, `BC` performs poorly. As our theory predicts, when we equip the `BC` learner with history, we actually see it perform *worse* on average, exhibiting the latching behavior that has been observed repeatedly in practice. In contrast, we see that equipping our on-policy learner with access to the last few observations and actions does not lead to a sharp decline in terms of performance. We use MSE as a surrogate for moment-matching error. We see that on both environments, on-policy methods are far better at matching expert actions on their own rollout distribution. We also see that with enough time, `DAgger-SEQ` is able to predict actions better than `DAgger` (lower $\epsilon_{\mathrm{on}}(1000)$). Putting together these observations, it appears as though on-policy methods with access to history are the most robust against partial information, agreeing with our theory. We release our code at `https://github.com/gkswamy98/sequence_model_il`.

# 7 Conclusion

We study asymptotically realizable imitation learning problems where the learner's ability to mimic the expert increases over the timesteps of the problem. We find that under identifiability and recoverability assumptions, on-policy algorithms are able to match time-averaged expert value, in contrast to off-policy algorithms which might pay for their early mistakes for the rest of the horizon. We demonstrate that empirically, off-policy algorithms are not able to take advantage of history to uncover missing information. We believe our results point towards a unified explanation of both empirical successes and failures.

# 8 Acknowledgments

ZSW is supported in part by the NSF FAI Award #1939606, a Google Faculty Research Award, a J.P. Morgan Faculty Award, a Facebook Research Award, an Okawa Foundation Research Grant, and a Mozilla Research Grant. GS is supported computationally by a GPU award from NVIDIA and emotionally by his family and friends.

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
