# A Proofs

*Proof of Theorem 5.4.* We proceed in cases. For concision, we write $f_t$ for $f(s_t, a_t, c)$ and $\tilde{f}_t$ for $\tilde{f}(h_t, a_t)$, where $(f, \tilde{f})$ are the pairs defined in Assumption 5.1.

**Online/Reward-matching.** By the definition of the value function, we can write that

$$\frac{1}{T}(J(\pi^E) - J(\pi)) = \frac{1}{T} \sum_{t=1}^{T} \mathbb{E}_{\tau \sim \pi^E}[r(s_t, a_t, c)] - \mathbb{E}_{\tau \sim \pi}[r(s_t, a_t, c)] \tag{21}$$

$$\leq \sup_{(f,\tilde{f}) \in \mathcal{F}_r \times \tilde{\mathcal{F}}_r} \frac{1}{T} \sum_{t=1}^{T} \mathbb{E}_{\tau \sim \pi^E}[f_t - \tilde{f}_t + \tilde{f}_t] - \mathbb{E}_{\tau \sim \pi}[f_t - \tilde{f}_t + \tilde{f}_t] \tag{22}$$

$$\leq \frac{1}{T} \sum_{t=1}^{T} \epsilon_{\text{rew}}(t) + \sup_{(f,\tilde{f}) \in \mathcal{F}_r \times \tilde{\mathcal{F}}_r} \frac{1}{T} \sum_{t=1}^{T} \mathbb{E}_{\tau \sim \pi^E}[f_t - \tilde{f}_t] - \mathbb{E}_{\tau \sim \pi}[f_t - \tilde{f}_t] \tag{23}$$

$$= \frac{1}{T} \sum_{t=1}^{T} \epsilon_{\text{rew}}(t) + \delta_{\text{rew}}(t). \tag{24}$$

Note that via Assumption 5.1,

$$\lim_{T \to \infty} \frac{1}{T} \sum_{t=1}^{T} \delta_{\text{rew}}(t) = 0. \tag{25}$$

We therefore can drop the latter term from our bound. By the definition of the lim sup, we know that $\forall \epsilon > 0, \exists T(\epsilon)$ s.t. $\forall t \geq T(\epsilon), \epsilon_{\text{rew}}(t) \leq \epsilon_{\text{rew}} + \epsilon$. Let

$$S(\epsilon) = \sum_{t=1}^{T(\epsilon)} \epsilon_{\text{rew}}(t) \tag{26}$$

denote the prefix sum. Then, we know that $\forall T' \geq T(\epsilon)$,

$$\sum_{t=1}^{T'} \epsilon_{\text{rew}}(t) = S(\epsilon) + \sum_{t=T(\epsilon)}^{T'} \epsilon_{\text{rew}}(t) \leq S(\epsilon) + (T' - T(\epsilon) + 1)(\epsilon_{\text{rew}}(\infty) + \epsilon). \tag{27}$$

Taking the average by dividing both sides by $T'$, we arrive at

$$\frac{1}{T'} \sum_{t=1}^{T'} \epsilon_{\text{rew}}(t) \leq \frac{S(\epsilon)}{T'} + (1 - \frac{T(\epsilon) - 1}{T'})(\epsilon_{\text{rew}}(\infty) + \epsilon). \tag{28}$$

Taking $\lim_{T' \to \infty}$ tells us that averages converge to at most $\epsilon_{\text{rew}}(\infty) + \epsilon$. Because this condition holds for all $\epsilon > 0$, we can take the $\lim_{\epsilon \to 0}$ to prove that

$$\lim_{T' \to \infty} \frac{1}{T'}(J(\pi^E) - J(\pi)) \leq \epsilon_{\text{rew}}(\infty). \tag{29}$$

**Interactive/On-Q.** We proceed similarly to the previous case. Via the Performance Difference Lemma [Kakade and Langford, 2002], we can write that

$$\frac{1}{T}(J(\pi^E) - J(\pi)) = \frac{1}{T} \sum_{t=1}^{T} \mathbb{E}_{\tau \sim \pi}[Q^{\pi^E}(s_t, a_t, c) - \mathbb{E}_{a \sim \pi^E}[Q^{\pi^E}(s_t, a, c)]] \tag{30}$$

$$\leq \sup_{(f,\tilde{f}) \in \mathcal{F}_{Q_E} \times \tilde{\mathcal{F}}_{Q_E}} \frac{1}{T} \sum_{t=1}^{T} \mathbb{E}_{\tau \sim \pi}[f_t - \tilde{f}_t + \tilde{f}_t - \mathbb{E}_{a \sim \pi^E}[f_t - \tilde{f}_t + \tilde{f}_t]] \tag{31}$$

$$\leq \frac{H}{T} \sum_{t=1}^{T} \epsilon_{\text{on}}(t) + \sup_{(f,\tilde{f}) \in \mathcal{F}_{\text{on}} \times \tilde{\mathcal{F}}_{\text{on}}} \frac{H}{T} \sum_{t=1}^{T} \mathbb{E}_{\tau \sim \pi}[f_t - \tilde{f}_t - \mathbb{E}_{a \sim \pi^E}[f_t - \tilde{f}_t]]$$

$$= \frac{H}{T} \sum_{t=1}^{T} \epsilon_{\text{on}}(t) + \delta_{\text{on}}(t). \tag{32}$$

The $H$ factor comes from the scaling of $\mathcal{F}_{\text{on}} = \{f/2H : f \in \mathcal{F}_{Q_E}\}$. As before, via Assumption 5.1,

$$\lim_{T\to\infty} \frac{1}{T} \sum_{t=1}^{T} \delta_{\text{on}}(t) = 0. \tag{33}$$

By the definition of the $\lim \sup$, we know that $\forall \epsilon > 0, \exists T(\epsilon)$ s.t. $\forall t \geq T(\epsilon), \epsilon_{\text{on}}(t) \leq \epsilon_{\text{on}} + \epsilon$. Let

$$S(\epsilon) = \sum_{t=1}^{T(\epsilon)} \epsilon_{\text{on}}(t) \tag{34}$$

denote the prefix sum. Then, we know that $\forall T' \geq T(\epsilon)$,

$$\sum_{t=1}^{T'} \epsilon_{\text{on}}(t) = S(\epsilon) + \sum_{t=T(\epsilon)}^{T'} \epsilon_{\text{on}}(t) \leq S(\epsilon) + (T' - T(\epsilon) + 1)(\epsilon_{\text{on}}(\infty) + \epsilon). \tag{35}$$

Taking the average by dividing both sides by $T'$, we arrive at

$$\frac{1}{T'} \sum_{t=1}^{T'} \epsilon_{\text{on}}(t) \leq \frac{S(\epsilon)}{T'} + (1 - \frac{T(\epsilon) - 1}{T'})(\epsilon_{\text{on}}(\infty) + \epsilon). \tag{36}$$

Taking $\lim_{T'\to\infty}$ tells us that averages converge to at most $\epsilon_{\text{on}}(\infty) + \epsilon$. Because this condition holds for all $\epsilon > 0$, we can take the $\lim_{\epsilon\to 0}$ to prove that

$$\lim_{T'\to\infty} \frac{1}{T'}(J(\pi^E) - J(\pi)) \leq H\epsilon_{\text{on}}(\infty). \tag{37}$$

**Offline/Off-Q.** Via the Performance Difference Lemma [Kakade and Langford, 2002], we can write that

$$\frac{1}{T}(J(\pi^E) - J(\pi)) = \frac{1}{T} \sum_{t=1}^{T} \mathbb{E}_{\tau\sim\pi^E}[Q^\pi(s_t, a_t, c) - \mathbb{E}_{a\sim\pi^E}[Q^\pi(s_t, a, c)]] \tag{38}$$

$$\leq \sup_{(f,\tilde{f})\in\mathcal{F}_Q\times\tilde{\mathcal{F}}_Q} \frac{1}{T} \sum_{t=1}^{T} \mathbb{E}_{\tau\sim\pi^E}[f_t - \tilde{f}_t + \tilde{f}_t - \mathbb{E}_{a\sim\pi}[f_t - \tilde{f}_t + \tilde{f}_t]] \tag{39}$$

$$\leq \frac{T}{T} \sum_{t=1}^{T} \epsilon_{\text{off}}(t) + \sup_{(f,\tilde{f})\in\mathcal{F}_{\text{off}}\times\tilde{\mathcal{F}}_{\text{off}}} \frac{T}{T} \sum_{t=1}^{T} \mathbb{E}_{\tau\sim\pi^E}[f_t - \tilde{f}_t - \mathbb{E}_{a\sim\pi}[f_t - \tilde{f}_t]]$$

$$= \sum_{t=1}^{T} \epsilon_{\text{off}}(t) + \delta_{\text{off}}(t). \tag{40}$$

The $T$ factor comes from the scaling of $\mathcal{F}_{\text{off}} = \{f/2T : f \in \mathcal{F}_Q\}$. Thus, we can write that

$$\lim_{T\to\infty} \sum_{t=1}^{T} \epsilon_{\text{off}}(t) + \delta_{\text{off}}(t) = \Sigma_{\text{off}}(\infty), \tag{41}$$

which implies that

$$\lim_{T\to\infty} \frac{1}{T}(J(\pi^E) - J(\pi)) \leq \Sigma_{\text{off}}(\infty). \tag{42}$$

$\square$

*Proof of Theorem 5.5.* Consider the Cliff problem of Swamy et al. [2021]. There is no hidden context in this problem so $\delta_{\text{off}}(t) = 0$. Let the learner take the action that puts them at the bottom of the cliff at timestep $t$ with probability $\frac{1}{t+1}$, giving us $\epsilon_{\text{off}}(t) = \frac{1}{t+1}$. Note that $\epsilon_{\text{off}}(t)$ decays to 0 but $\Sigma_{\text{off}}(t)$ does not as the harmonic series diverges. Once the learner falls off the cliff, they recieve no reward for the rest of the horizon. This means that

$$\frac{1}{T}(J(\pi^E) - J(\pi)) = \frac{1}{T} \sum_{t=1}^{T} \frac{T-t}{t+1} = \sum_{t=1}^{T} \frac{1}{t+1}(1 - \frac{t}{T}) = \sum_{t=1}^{T} \epsilon_{\text{off}}(t)(1 - \frac{t}{T}). \tag{43}$$

The limit of the sum of the first term is $\Sigma_{\text{off}}(\infty)$. For the second term,

$$\lim_{T \to \infty} \frac{1}{T} \sum_{t=1}^{T} \frac{t}{t+1} = 1. \tag{44}$$

Thus,

$$\lim_{T \to \infty} \frac{1}{T}(J(\pi^E) - J(\pi)) = \Sigma_{\text{off}}(\infty) - 1 \gtrsim \Sigma_{\text{off}}(\infty). \tag{45}$$

$\square$

*Proof of Theorem 5.7.* Define the infinite-horizon payoffs [2] for our moment-matching games as follows:

$$U_{\text{rew}}(\pi, f) = \lim_{T \to \infty} \mathbb{E}_{\tau \sim \pi}[f(h_T, a_T)] - \mathbb{E}_{\tau \sim \pi^E}[f(h_T, a_T)], \tag{46}$$

$$U_{\text{on}}(\pi, f) = \lim_{T \to \infty} \mathbb{E}_{\tau \sim \pi}[f(h_T, a_T)] - \mathbb{E}_{\tau \sim \pi, a \sim \pi_E}[f(h_T, a)]. \tag{47}$$

Note that under Asymptotic Realizability (Assumption 5.1), there exists a policy $\pi \in \Pi$ s.t. $\forall f \in \tilde{\mathcal{F}}$, $U_{\text{rew}}(\pi, f) = 0$ and $U_{\text{on}}(\pi, f) = 0$.

Let $\pi_{\text{rew}}$ and $\pi_{\text{on}}$ denote $\epsilon$-approximate Nash equilibrium strategies for the above two games (which could be computed by, say, running a no-regret algorithm over $\Pi$ against a no-regret or best-response counterpart for the $f$ player). By the definition of an approximate Nash equilibrium, we know that

$$\sup_{f \in \tilde{\mathcal{F}}_r} U_{\text{rew}}(\pi_{\text{rew}}, f) - \epsilon \leq \inf_{\pi \in \Pi} U_{\text{rew}}(\pi, f) = 0, \tag{48}$$

where the last step comes from our realizability assumption. This implies that

$$\sup_{f \in \tilde{\mathcal{F}}_r} U_{\text{rew}}(\pi_{\text{rew}}, f) = \epsilon_{\text{rew}}(\infty) \leq \epsilon. \tag{49}$$

Similarly, we can write that

$$\sup_{f \in \tilde{\mathcal{F}}_{\text{on}}} U_{\text{on}}(\pi_{\text{on}}, f) = \epsilon_{\text{on}}(\infty) \leq \epsilon. \tag{50}$$

Plugging these expressions into Theorem 5.4 gives us the desired results.

$\square$

*Proof of Corollary 5.8.* Assume the learner is subject to an $\epsilon_{exp} > 0$ probability of playing a different action than intended (either as part of the dynamics or because of explicit exploration noise). Consider the following function:

$$\tilde{f}(h_t, a_t) = \mathbf{1}[a_t = \max_k^K \frac{n_k^+}{n_k}], \tag{51}$$

where $n_k$ refers to the total number of pulls of arm $k$ and $n_k^+$ refers to the number of pulls of arm $k$ that elicit positive feedback. We proceed by arguing that this function will converge to the reward function of the problem. We specialize on the two-arm case as it is the most difficult for the learner. W.l.o.g., let arm 1 be the correct arm. Note that $r_1 = \frac{n_1^+}{n_1}$ and $r_2 = \frac{n_2^+}{n_2}$ are both averages of Bernoulli coin flips. Thus, via a Hoeffding bound, we know that

$$P(r_2 \geq r_1) = P(r_2 - \mathbb{E}[r_2] \geq r_1 - \mathbb{E}[r_2]) \tag{52}$$

$$= P(r_2 - \epsilon_{obs} \geq r_1 - \epsilon_{obs}) \tag{53}$$

$$\leq \exp\left(\frac{-2(r_1 - \epsilon_{obs})^2}{n_2}\right) = \delta(t). \tag{54}$$

Given that $(r_1 - \epsilon_{obs})^2$ is bounded and w.h.p. not equal to 0, we can say that $\lim_{t \to \infty} \delta(t) = 0$ as $\lim_{t \to \infty} n_2 = \infty$ because of the exploration noise / dynamics. Thus, we know that eventually, $r_1 < r_2$, which implies that $\tilde{f}(h_t, a_t) = \mathbf{1}[a_t = 1]$, which is the reward function of the problem. This

---

[2] When this limit exists, the average over timesteps of moment-matching error is equal to it.

means that we are asymptotically reward-moment identifiable for the minimal reward-moment class, $\mathcal{F}_r = \{r\}$. As we made no restrictions on the action distribution for this problem, this means the problem is trivially realizable. Thus, by Theorem 5.4, matching this moment in an on-policy fashion is sufficient to achieve AVE.

$\square$

## B Experiments

### B.1 Causal Bandit Experiments

The results we present are with $K = 5$ and after $T = 2000$ timesteps averaged across 100 trials. We add explicit exploration noise in the form of an $\epsilon_{exp}$ chance of playing an arm other than the one the learner chose. We start off all learners with a uniform prior and check and see if at $t = T$ whether they pick the correct arm with probability at least $\epsilon_{exp} - 0.12$. If so, we add a green dot. Otherwise, we add a red dot. We refer interested readers to our code for the precise expressions we used but, roughly speaking, we perform Bayesian filering with or without treating the actions as evidence. As argued above, this corresponds to assuming the on-policy or off-policy graphical models of Fig. 2.

### B.2 PyBullet Experiments

We give the off-policy learners 25 demonstration trajectories, each of length 1000. As described above, our non-sequential models are MLPs with two hidden layers of size 256 and ReLu activations. Our sequential models are LSTMs with hidden size 256 followed by an MLP with one hidden layer of size 256. We use a history of length 5 for all experiments and train all learners with a MSE loss and an Adam optimizer [Kingma and Ba, 2014] with learning rate $3e - 4$. Our sequence models are given access to the last $5$ states and the last $4$ actions and are asked to predict the next action. We evaluate MSE and $J(\pi)$ by rolling out 100 trajectories and averaging.

**HalfCheetah Experiments.** As in Finn et al. [2017], we sample a target velocity for the agent from $U[0, 3]$, which is passed in as part of the state to the expert but hidden from the learner. We train an expert for this task via Soft Actor Critic (SAC) [Haarnoja et al., 2018] – we refer interested readers to our code for precise hyperparameters. The reward function we train the expert and evaluate learner policies with is

$$1 - |\dot{x}_t - c| - 0.05||u_t||_2^2, \tag{55}$$

where $c$ is the target velocity. We run behavioral cloning for $1e5$ steps. For DAgger [Ross and Bagnell, 2010], we train for $5e4$ steps on the same set of 25 trajectories as were given to the off-policy learners and then perform 9 iterations of rollouts/aggregation/refitting, sampling 20 trajectories and training for $5e3$ steps. Thus, both DAgger and BC are given the same compute budget – the only difference is the data that is passed in.

**Ant Experiments.** We sample a target velocity for the agent from $U[0, 1.5]$ and mask it for the first 200 timesteps and then reveal it to the learner. We train the expert policy using reward function

$$1 - |\dot{x}_t - c| - 0.5||u_t||_2^2, \tag{56}$$

where $c$ is the target velocity. We filter demonstrations to only include expert trajectories that have at least 500 environment steps. We use the same model classes as for HalfCheetah but add in dropout to the input with $p = 0.5$ for the sequence models as it helps uniformly. We run behavioral cloning for $1e5$ steps. For DAgger [Ross and Bagnell, 2010], we train for $1e4$ steps on the same set of 25 trajectories as were given to the off-policy learners and then perform 9 iterations of rollouts/aggregation/refitting, sampling 25 trajectories and training for $1e4$ steps. Thus, both DAgger and BC are given the same compute budget – the only difference is the data that is passed in.