# OpenReview forum: "Sequence Model Imitation Learning with Unobserved Contexts"
_NeurIPS.cc/2022/Conference — NeurIPS 2022 Accept_

### Official Review · Reviewer_wNyh · 2022-07-06

**Rating:** 7
**Confidence:** 3
**Soundness:** 3 good
**Presentation:** 3 good
**Contribution:** 3 good

**Summary:**

This paper investigates the feasibility of performing imitation learning in a setting where the contexts (in the sense of Contextual MDPs) are unobserved. They show theoretically that purely offline imitation learning methods in this setting can fail to learn the correct causal policy, due to the unobserved context acting as a kind fo confounder. They show that online imitation learning methods which assume access to a queriable expert or environment can address this problem, and that in cases where the context can be identified from the transitions, and early failures aren't recoverable, then these methods can asymptotically match expert value. They perform several experiments on HalfCheetah where the context defines the target speed (i.e. the reward function), and demonstrate that online IL methods perform better than offline IL methods, as the theory predicts.

**Questions:**

- L292: We don't know that adding history is leading to latching-style behaviour - it could be because optimisation with an LSTM is harder - how much did you sweep the LSTM HPs and architecture?
- L294: Shouldn't adding history improve the on-policy learner's performance, and is history is necessary for realisability?
- Fig 4 left - could you plot the expert's performance here for comparison?
- Fig 4 mid and right - Could you be more clear how you're calculating the y-axis values?
- L221: It would be beneficial to redefine F_Q_E, perhaps in appendix, so the paper is self-contained.


**Limitations:**

I think the limitations are sufficiently described in the paper

**Strengths And Weaknesses:**

Overall I like the paper, and I think it provides some clear and intuitive conditions for when offline IL may fail, and how online IL can address these failures. The paper is clearly written, and the contribution is novel. The well-argued theoretical results, along with the conclusions of the results, are the main strengths of the paper.

I think the main weakness of the paper lies in the lack of experimental validation of the theoretical results. While I don't think this suffices as a weakness to reject the paper, I think it could be substantially improved with more experimental evidence that aligns with the theoretical conclusions.

For example, performing experiments on multiple environments would improve the robustness of any conclusions drawn from the experiments. Further, the theory would seem to suggest that having access to history makes the task realisable, meaning that the policy should have improved performance, but this isn't the case in the experiments. Experiments with MDPs where you can adjust whether the realisability and identifiability assumptions are satisfied and investigating how that affects the performance of offline and online IL methods would be beneficial.

The best version of these experiments would be a set of experiments across multiple environments, where it's shown that satisfying the assumptions improves the performance of the online IL algorithms more than it improves the performance of the offline IL algorithms (if they improve at all), and that metrics that the theory predicts to behave in certain ways behave in those way empirically. I realise this is a big ask, and as I said I don't think it's necessary for the paper to be accepted, but it would improve the robustness of the scientific contribution of this paper.

As a separate point, it would be great to see examples of real-world examples that satisfy the asymptotic identifiability and realisability assumptions, and those that don't, to get a clearer intuition of when we can expect these methods to succeed in practice.

---

> ### Author Response · Authors · 2022-08-02
> **Re:**
>
> We appreciate the reviewer’s enthusiasm for our work – we are also quite excited about the possibilities of online IL for reducing the effect of missing information. We would like to respond to the concerns raised.
>
> Experiments: As requested, we added an additional experiment on the AntBulletEnvironment where the target speed is masked from the learner for the first 200 timesteps but is always visible to the expert. We see similar patterns here with respect to the performance of the four methods at matching expert actions and total reward.
>
> Q1: We swept the number of layers in the LSTM and found negligible improvements. We note that the on-policy sequence model trains fine, indicating that the issue here is one of the data being passed in rather than the model class or the optimization procedure, as the data is the only thing that differs between the on and off-policy setups.
>
> Q2: This is a great question! The middle plot of Fig. 4 shows us that the sequence models are able to (asymptotically) drive down action MSE w.r.t the expert policy more than their non-sequential counterparts. This means that there exists a reasonable reward function (match the mean expert action) on which the sequence models perform better. However, on the actual reward function of the problem, the non-sequential on-policy method happens to perform as well as the sequential method. For the HalfCheetah problem, history is required for realizability as the learner can otherwise only determine if they have to go faster or slower but not by how much. For the Ant problem, it is not as the context is eventually revealed to the learner. In both settings however, we see adding history hampers the performance of off-policy methods as they treat their own actions early on though they came from the expert.
>
> Q3: The expert achieves 300 for Ant and 560 for HalfCheetah, which we added to caption of Fig. 4. We emphasize that the relatively low total rewards here are because the expert is solving a multi-task reinforcement learning problem (in contrast to the standard Bullet tasks which are far easier). There is a performance gap between DAgger and the expert as we consider total reward rather than asymptotic average reward. This is as expected because the expert sees the context from the beginning of the episode while the learner has to interact with the environment to try and narrow it down. While our theory predicts the performance gap between the learner and expert would vanish on average asymptotically, because we consider finite-horizon problems (H = 1000), we still see some.
>
> Q4: We were calculating the MSE between learner and expert actions on learner (mid) and expert (right) trajectories. We removed the right plot as we found it less relevant than the performance of the learner under its own induced state visitation distribution.
>
> Q5: We added in a new section (5.1) providing a background on moment-matching in imitation learning which should hopefully resolve some confusion.

---

### Official Review · Reviewer_GRay · 2022-07-12

**Rating:** 6
**Confidence:** 3
**Soundness:** 3 good
**Presentation:** 2 fair
**Contribution:** 3 good

**Summary:**

The paper discusses the performance bound of on-policy and off-policy imitation learners and the conditions for the learners to achieve expert performance. The theoretical results show that on-policy learners have stronger guarantees than off-policy learners. Experimental results also support the conclusion.

**Questions:**

For the experiment results in figure 4:
1. In the left plot, both learners' performances become worse after adding history information. Can we have a more detailed explanation of this?
2. In the middle plot, the -SEQ methods are all better than their original counterparts, which are opposite to the resuts we see in the left plot. Why is this?
3. In the middle plot, while the final performance of DAGGER-SEQ is better than DAGGER, it seems to have a worse sample efficiency. Can we have some discussion of this?
4. For the right plot, can we also have an experiment that shows when BC is able to achieve value equivalance?

**Limitations:**

Please consider adding more explanation to the questions above.

**Strengths And Weaknesses:**

Strengths:
1. The paper provides both theoretical proofs and empirical results.
2. The theoretical results are intuitive and are given with practical conditions for achieving desired performance (Assumption 3.6 and 3.7).

Weaknesses:
The experiment results are a bit weak and incomplete. Some of the results are lack of explaination.

---

> ### Author Response · Authors · 2022-08-02
> **Re:**
>
> We thank the reviewer for their comments and would like to attempt to answer their questions. In order:
>
> 1. We see a noticeable drop in the performance of BC after adding history, which both mirrors real-world experiments and matches with what our theory predicts. In short, because an off-policy learner often learns to simply repeat previous actions (i.e. the latching effect), it can perform quite poorly on rollouts as it repeats its initial random action. DAgger performs comparably with and without a sequence model policy – the standard error bars overlap.
> 2. This is an excellent observation that gets at the core of our insight! Assuming a powerful enough model class, giving a learner access to history (i.e. more information) increases their ability to drive down moment matching error, which is what we see in the middle plot. Our theory argues that even when an off-policy learner is able to drive down moment-matching error, they are not able to take advantage of this fact – this is why we don’t have a guarantee that the off-policy sequential model will perform better than the regular off-policy method in terms of total reward. We added in an example (Thm. 5.5) when this is provably not the case. The fact that the on-policy sequence method doesn’t perform better than the non-sequential version is a product of the reward function for the problem: if the learner were more heavily penalized for deviating from the behavior of the expert, the sequence model variant would perform better. With the additional page provided for later versions, we would be happy to add more discussion of this point.
> 3. We believe this reflects a tradeoff between model flexibility and degree of covariate shift. Asymptotically, because it has more parameters, one would expect the sequence model to perform better. However, early on in an episode there is limited data and therefore a relatively large amount of covariate shift (as the expert policy is not yet realizable). This is especially true for the sequence model as it depends on more elements of the past and therefore is more vulnerable to covariate shift – i.e., there is a tighter feedback loop because of more dependence on the past. Put differently, the tradeoff is between how helpful a sequence is for narrowing down the context and the fact that a more powerful model requires more samples.
> 4. On the full-information version of these problems, BC is able to match expert performance (see for example the works of Swamy et al. or Spencer et al.). This is inconsistent with the real world, where BC often performs poorly [1]. Thus, we introduce hidden contexts into our problems to attempt to capture a facet of what makes the imitation learning problem hard in practice – we therefore wouldn’t expect BC to achieve value equivalence to the expert policy.
>
> [1] https://proceedings.neurips.cc/paper/1988/file/812b4ba287f5ee0bc9d43bbf5bbe87fb-Paper.pdf

---

> ### Author Response · Authors · 2022-08-09
> **Re:**
>
> Hello! As the author-reviewer discussion period is coming to a close, please let me know if any addition clarifications (in addition to those presented in our previous response) would be helpful in your evaluation.

---

### Official Review · Reviewer_Pth8 · 2022-07-13

**Rating:** 3
**Confidence:** 3
**Soundness:** 2 fair
**Presentation:** 1 poor
**Contribution:** 1 poor

**Summary:**

In this paper, the authors study imitation learning in contextual Markov decision processes (CMDPs). Three different frameworks, off-policy, online and interactive, are considered. The authors inspect the differences among the frameworks with structural causal models to show the issue of off-policy methods, such as behavioral cloning, Theoretical analyses for performance bounds are provided as well as a short empirical evaluation.

**Questions:**

1. Line 177: Bayes rule implies that $p(c|h_t) = p(c, h_t) / p(h_t)$. $p(h_t) = \sum_c p(h_t|c)p(c)$. Even with a uniform prior over contexts, $p(h_t)$ will not have a uniform distribution. By writing equation (3), $p(c|h_t) \propto p(c, h_t)$, the authors imply we could ignore $p(h_t)$. Am I missing something here?


2. Line 185: in equation (6), $\mathcal{G}$ is not defined.


3. Line 222: I didn’t understand what “$\mathcal{F}_{Q_E}$ spans the set of possible expert Q-functions” means. Given an expert policy $\pi_E$, shouldn’t there just be a corresponding Q function $Q^{\pi_E}$?


4. In section 4, can the authors provide what the expert’s performance is, i.e., $J(\pi_E)$? Is there a performance gap in the DAgger methods?


**Limitations:**

The authors had an adequate discussion on the limitation. I think analyzing the finite-sample setting will be very interesting.

**Strengths And Weaknesses:**

Originality: I found the novelty of this paper to be limited. The main difference from the work by Swamy et al. is the introduction of contexts in MDPs. However, the section (section 3.2) connecting CMDP with structural causal models needs improvement. I am not an expert in causal learning, but after attempting to read this section multiple times, I still could not understand what the connection is.

In terms of the theoretical results, they largely mirror those in Swamy et al. as well. In fact, the inclusion of the context doesn’t introduce anything new in the analysis. So I didn’t see much novelty here either.

Finally, the CMDP-specific result, Theorem 3.8, is interesting, but due to the strong assumption, Assumption 3.6, the value of the theorem is limited.

Overall, studying performance gap bounds in CMDP is interesting, but the results in the paper is not novel enough.

Quality and clarity: I think the paper needs a significant rewrite to improve its quality and clarity. Many important details are left out which makes the paper confusing to read.

1. In the bandit example in section 3.1, the authors could explain in more detail what applying DAgger in a bandit setting looks like. Moreover, it is unclear what form the policy is, is it a linear model, a tabular agent, or something else? The appendix B.1 failed to answer these basic questions as well.

2. Section 3.2 assumes the reader is familiar with causal learning, such as the part from line 184 - 186. I think many in the imitation learning community will appreciate relevant concepts explained as preliminary before the main message. The same goes for the use of structural causal models (Figures 2 and 3), how does one interpret the color and direction of an arrow?

3. I list some technical questions in the next section as well, which could improve the clarity of the paper.

Significance: I think the topic is relevant to the community but the results in the paper is not significant to clear the bar for NeurIPS.

---

> ### Author Response · Authors · 2022-08-02
> **Re:**
>
> We thank the reviewer for their comments and suggestions and have revised our draft in response (see below). We would like to respond to several concerns raised:
>
> Originality: We believe the core contribution of our work is the consideration of settings in which the learner can more closely mimic expert behavior as they accumulate experience across an episode. We prove a new, dramatic separation between on-policy (e.g. GAIL, DAgger) and off-policy (e.g. behavioral cloning) algorithms on such problems that we haven’t seen elsewhere in the literature and that captures a situation common in practice.
>
> We have re-written parts of the SCM section to try and make it more clear. The core point of this section is that off-policy algorithms are in a certain sense performing inference in a causal model (Fig. 2, c) that differs from reality (Fig 2, b). By treating their own past actions as interventions rather than as evidence about the missing context, on-policy methods do not make this issue of causation that leads to a latching effect.
>
> Theoretical Results: Our analysis utilizes an asymptotic notion of moment-identifiability which differs from the prior work of Swamy et al. (who focus on problems where the learner can, from the beginning of an episode, imitate the expert well). The key result of our work is that even when this identifiability condition holds true, off-policy methods aren’t able to take advantage of it. This is because of the facts that (1) early on in an episode, all methods are guaranteed to make mistakes. However, because (2) off-policy methods are only trained on data from the expert’s state visitation distribution, they don’t learn to recover from these unavoidable mistakes. In a sense, our results show how partial observability (a special case of asymptotic realizability) can be seen as a source of early-episode mistakes that compound over time and lead to poor performance over the horizon. We have re-written our results in terms of a weaker assumption to try and emphasize the novelty of our point.
>
> 1) We apologize for the lack of clarity here – we use DAgger as a stand-in term for an on-policy algorithm. For the presented experiments, we maintain explicit posteriors over the context (i.e. the best arm) and then follow the policies given in eqs. 7/8. On an implementation-level detail, given there are a finite number of contexts, we maintain a tabular representation of this posterior. Our provided code gives the precise expressions we used to perform the posterior updates.
>
> 2) We use color to highlight whether the learner or expert is taking actions. The tail of an arrow is an element in the head’s conditional probability distribution. We tried to add more visuals to support the graphical models.
>
> Q1: We dropped the denominator to make a point about the ratio between on and off-policy context posteriors. Fix an arbitrary history. Then, up to a constant factor, the ratio of the on-policy and off-policy posteriors will be the product of expert policy action probabilities. The reviewer is correct that the constant factor will depend on the particular history but we were trying to draw attention to the product of expert action probabilities term.
>
> Q2: G is the on-policy graphical model defined in Figure 2, (b). The notation is borrowed from Pearl (see for example his 3 rules in https://ftp.cs.ucla.edu/pub/stat_ser/r236-3ed.pdf).
>
> Q3: We added in a new section (5.1) that will hopefully provide a better background on moment-matching in imitation learning. The reviewer is correct that there exists a single Q_E in reality. If we knew this function, we would be able to perform as well as the expert by simply taking the argmax over actions. However, in imitation learning, we often do not know the ground-truth reward function and therefore do not know the Q_E function. The solution proposed by Abbeel and Ng and expanded upon by Ziebart et al. is to consider a class of reward functions that is assumed to contain the true reward function and ensure the learner matches expert performance under all of these functions. Swamy et al. argue that off-policy algorithms are performing an analogous operation over potential Q functions of the expert policy under different rewards.
>
> Q4: Sure! The expert achieves 300 for Ant and 560 for HalfCheetah. We emphasize that the relatively low total rewards here are because the expert is solving a multi-task reinforcement learning problem (in contrast to the standard Bullet tasks which are far easier). There is a performance gap between DAgger and the expert as we consider total reward rather than asymptotic average reward. This is as expected because the expert sees the context from the beginning of the episode while the learner has to interact with the environment to narrow it down. While our theory predicts the performance gap between the learner and expert would vanish on average asymptotically, because we consider finite-horizon problems (H = 1000), we still see some.

---

> > ### Comment · Reviewer_Pth8 · 2022-08-09
> > **Thanks for your response**
> >
> > Thanks for your response! For the novelty, in terms of the performance gap bounds for DAgger and behavioral cloning, how are they different from those in earlier works [1, 2]?
> >
> > 1. Ross, Stéphane, and Drew Bagnell. "Efficient reductions for imitation learning." Proceedings of the thirteenth international conference on artificial intelligence and statistics. JMLR Workshop and Conference Proceedings, 2010.
> > 2. Ross, Stéphane, Geoffrey Gordon, and Drew Bagnell. "A reduction of imitation learning and structured prediction to no-regret online learning." Proceedings of the fourteenth international conference on artificial intelligence and statistics. JMLR Workshop and Conference Proceedings, 2011.

---

> > > ### Author Response · Authors · 2022-08-09
> > > **Re:**
> > >
> > > They differ in a few ways:
> > >
> > > 1) The bounds we derive here are on the entire classes of offline (e.g. BC) and interactive (e.g. DAgger) imitation learning algorithms, not just those two algorithms as in [1, 2], giving us the ability to make significantly more general claims. We do this by taking the perspective of moment-matching (building upon Swamy et al.'s work).
> > >
> > > 2) The bounds we derive here allow for reward functions that depend on unobserved contexts, while the DAgger-style proofs or those of Swamy et al. are unable to handle this challenge. More generally, any version of partial observability on the part of the learner that fits under the identifiability conditions we propose could be handled by the sort of arguments we make in this paper.
> > >
> > > 3) A bit more technically, the on-policy bounds we derive here depend on lim-sups of moment-matching errors, while the DAgger-style / Swamy et al. -style bounds depend on average-over-timestep errors (e.g. the $\epsilon$ in the DAgger proofs is the average error).

---

> ### Author Response · Authors · 2022-08-08
> **Re:**
>
> Hello! We believe we have addressed some of the concerns you brought up in your review. Please let us know if there's any other information we could provide that would help with your evaluation.

---

### Meta-Review · Area_Chair_K1yW · 2022-08-28

**Recommendation:** Accept
**Confidence:** Certain

**Metareview:**

The paper proposes a new imitation learning setting where some context known by the expert is unobserved by the learner (both during learning and when exploiting the learnt policy). The value of this contribution has been acknowledged by the reviewers.

Some reviewers raised questions about the novelty of this work compared to specific existing work. The authors provided fair responses that seemed convincing.

Some reviewers were also concerned by the quality of the experimental evaluation. The authors proposed a new experiment. It may not fully address the concerns of the reviewers but it seems that the value of the theoretical contribution combined with these experiments is enough for being considered for publication.


**Award:**

No

---

### Decision · Program_Chairs · 2022-09-14

Accept